# DINO: A Diabatic Model of Pole-to-Pole Ocean Dynamics to Assess Subgrid Parameterizations across Horizontal Scales

David Kamm<sup>1,2</sup>, Julie Deshayes<sup>1,3</sup>, and Gurvan Madec<sup>1,4</sup>

Correspondence: David Kamm (david.kamm@locean.ipsl.fr)

Abstract. Climate models are limited in resolution by computational constraints. The ocean component is currently resolved at spatial scales between approximately 10 to  $100 \, \mathrm{km}$ , which is too coarse to adequately capture the mesoscale. Eddies at these scales play a major role in the global energy cycle, and therefore it is crucial that they are accurately parameterized. In this context, we propose DINO (*DIabatic Neverworld Ocean*), an ocean-only model configuration of intermediate complexity designed as a test protocol for eddy parameterizations across a range of horizontal scales. It allows for affordable simulations, even at very high resolution, while crucial aspects of the global ocean like the Meridional Overturning Circulation (MOC), Subtropical and Subpolar gyres, or the Antarctic Circumpolar Current (ACC) are maintained. We compare key metrics across eddy-resolving  $(1/16^{\circ})$ , eddy-permitting  $(1/4^{\circ})$  and eddy parameterizing  $(1^{\circ})$  simulations to showcase the evaluation of eddy parameterizations in two ways: by testing their impact on the mean state and by directly diagnosing the missing eddy fluxes from coarse-grained high-resolution experiments.

#### 1 Introduction

20

The vast range of spatial and temporal scales of Earth's climate system and the underlying processes involved make numerical climate simulations a computationally costly endeavor: it requires representing the effect of small scales ( $< 100 \, \mathrm{km}$ ) in long simulations ( $> 500 \, \mathrm{years}$ ). Limited available computational resources therefore impose constraints on the horizontal resolution of future climate projections. At the time of writing this article, the ocean component of such models ranges in grid size between approximately  $10 \, \mathrm{km}$  and  $100 \, \mathrm{km}$  (Hewitt et al., 2020). These grid scales coincide with the horizontal scale of geostrophic turbulence (Chelton et al., 1998). A prominent feature associated with turbulence at these scales is the formation of mesoscale eddies. They are the dominant reservoir of kinetic energy (KE) in the ocean and play a key role in its energy cycle (Ferrari and Wunsch, 2009).

Winds inject kinetic energy at the surface, and, along with heterogeneous buoyancy forcing, they sustain a reservoir of potential energy (PE) at large scales. The thereby excited baroclinic modes and nonlinear interactions between them lead to energy transfers across scales. (Charney, 1971). Upon reaching scales close to the deformation radius of the first mode, baroclinic instabilities drive the formation of eddies, thereby converting available potential energy (APE) to eddy kinetic

<sup>&</sup>lt;sup>1</sup>Sorbonne Université-CNRS-IRD-MNHN, LOCEAN Laboratory, Paris, France

<sup>&</sup>lt;sup>2</sup>Sorbonne-Université

<sup>&</sup>lt;sup>3</sup>Sorbonne Université, Université Paris Cité, CNRS, INRIA, LJLL, EPC ANGE, F-75005 Paris, France

<sup>&</sup>lt;sup>4</sup>Univ. Grenoble Alpes, INRIA, CNRS, Grenoble INP, LJK, EPC AIRSEA, Grenoble, France

energy (EKE). Turbulence theory predicts an inverse energy cascade, which then transfers KE back to larger scales (Rhines, 1977).

When numerical models do not resolve mesoscale eddies explicitly nor parameterize them accurately, APE is not sufficiently converted to EKE, which in turn is not transferred to the larger scale KE budget. This leads to biases in the model's mean state and variability, adding uncertainty to the future response of climate projections (Hewitt et al., 2020). As computational cost limits the explicit representation of eddies in the global oceans, substantial efforts of the ocean modeling community have been dedicated to the development of eddy parameterizations over the last decades.

Gent and Mcwilliams (1990) (GM) introduced a parameterization for tracer transport driven by baroclinic instabilities. It diffuses the APE of the large-scale flow, leading to the flattening of isopycnal surfaces, consistent with the effects of resolved baroclinic instabilities. However, it does not account for the injection nor inverse cascade of kinetic energy. To address this, KE backscatter parameterizations have been developed to represent the inverse KE cascade from the subgrid scale when eddies are only partially resolved (Jansen and Held, 2014; Juricke et al., 2019; Eden and Greatbatch, 2008; Bachman, 2019).

Recent studies have used machine learning techniques to learn the missing eddy fluxes of momentum and buoyancy directly from high-resolution model data (Zanna and Bolton, 2020; Guillaumin and Zanna, 2021). Once a parameterization is implemented into an ocean general circulation model, it is not straightforward to evaluate the effect on the model solution given the limited availability of high-resolution reference simulations or observations. To address this challenge, we propose an idealized configuration of pole-to-pole ocean dynamics, a *Dlabatic Neverworld Ocean* (DINO). As the name suggests, it is broadly based on the Neverworld2 configuration (Marques et al. (2022); in the following NW2), but it accounts for diabatic processes such as convection, diapycnal mixing, and dense water formation.

Neverworld2 is strictly adiabatic, and for good reasons: the adjustment time is much shorter; the calculation of APE is straightforward; and ultimately the circulation is nearly adiabatic in most parts of the ocean in any case (Pedlosky, 1996). However, these benefits are not without trade-offs. The model does not include explicit tracer equations for temperature and salinity but a single tracer with a linear equation of state. The volume of water masses represented by each layer remains constant after initialization. Important metrics of the climate system, such as the meridional heat transport or the meridional overturning circulation, are not captured accurately with such a model. The introduced DINO configuration aims to complement Neverworld2 in these aspects but retains its overall objective of serving as a test case for the development and assessment of eddy parameterizations.

Section 2 describes the DINO configuration in detail, followed by its numerical implementation in Section 3. In Section 4, we outline the experiment design and present the results in Sections 5 and 6: a comparison of the mean state at eddy-parameterizing (1°) and eddy-permitting (1/4°) resolutions in Section 5, and an analysis of the missing subgrid momentum fluxes at eddy-permitting resolutions in Section 6. We discuss the findings in Section 7 and conclude in Section 8.

# 55 2 The DINO configuration

## 2.1 Model equations

We solve the primitive equations, which describe an incompressible fluid in a rotating system under the Boussinesq and hydrostatic approximation. The primary prognostic equations are given by the momentum balance for the zonal (u) and meridional (v) components of the velocity vector field  $u = u_h + kw$ .

60 
$$\partial_t \boldsymbol{u}_h + \left[ (\boldsymbol{\nabla} \times \boldsymbol{u}) \times \boldsymbol{u} + \frac{1}{2} \boldsymbol{\nabla} \boldsymbol{u}^2 \right]_h + f \boldsymbol{k} \times \boldsymbol{u}_h + \frac{1}{\rho_0} \boldsymbol{\nabla}_h p = \boldsymbol{D}^{\boldsymbol{u}} + \boldsymbol{F}^{\boldsymbol{u}}$$
 (1)

and the conservation equations for conservative temperature  $\Theta$  and absolute salinity  $S_A$ 

$$\partial_t \Theta + \nabla \cdot (\Theta \mathbf{u}) = D^{\Theta} + F^{\Theta} \tag{2}$$

$$\partial_t S_A + \nabla \cdot (S_A \mathbf{u}) = D^{S_A} + F^{S_A},\tag{3}$$

where p is the pressure, k is the unit vector in the vertical direction,  $F^{u/\Theta/S_A}$  represent forcing terms and  $D^{u/\Theta/S_A}$  potential parameterizations of small-scale physics for the corresponding prognostic variable. The subscript  $(...)_h$  denotes the local vector component in the horizontal plane. The remaining parameters are the Coriolis frequency f and a reference density  $\rho_0$ . The vertical velocity component f0 and the pressure f1 follow diagnostic equations assuming that the fluid is incompressible and in hydrostatic equilibrium, namely

$$\nabla \cdot \boldsymbol{u} = 0, \tag{4}$$

$$\partial_z p = -\rho g, \tag{5}$$

where g is the gravitational acceleration and the density  $\rho$  is a function of  $\Theta$ ,  $S_A$  and the depth z, defined by the equation of state (EOS) for seawater. We follow the formulation proposed by Roquet et al. (2015a) and approximate a simplified EOS as

$$\rho(\Theta, S_A, z) = \rho_{\text{ref}}(z) - \left(a_0 + \frac{1}{2}C_b\Theta_a + T_h z\right)\Theta_a + b_0 S_a,\tag{6}$$

where  $\Theta_a = \Theta - 10\,^{\circ}\mathrm{C}$  and  $S_a = S_A - 35\,\mathrm{g\,kg^{-1}}$ . We use the linear thermal expansion coefficient  $a_0$ , thermal cabbeling coefficient  $C_b$ , thermobaric coefficient  $T_h$  and linear haline expansion coefficient  $b_0$  following Caneill et al. (2022). The reference density profile  $\rho_{\mathrm{ref}}$  is arbitrary for a Boussinesq fluid, as only horizontal density gradients enter the governing equations through the pressure gradient term (Roquet et al., 2015b). We use characteristic values of the in situ density where absolute values are needed, for example  $\rho_{\mathrm{ref}}(z=0) = 1026\,\mathrm{kg\,m^{-3}}$  and  $\rho_{\mathrm{ref}}(z=2000) = 1035\,\mathrm{kg\,m^{-3}}$  to compute the potential density referenced to the surface or 2000 m depth. Numerical values of all constant parameters and their physical meaning can be found in Table 1.

**Table 1.** Numerical values, physical units and a short description of the constant parameters of the model.

| Parameter    | Value                  | Units                                       | Meaning                        |
|--------------|------------------------|---------------------------------------------|--------------------------------|
| $ ho_0$      | 1026                   | ${\rm kgm^{-3}}$                            | reference density              |
| $c_p$        | 3991.86                | $\rm Jkg^{-1}K^{-1}$                        | specific heat capacity         |
| $a_0$        | 0.165                  | ${\rm kg}{\rm m}^{-3}{\rm K}^{-1}$          | thermal expansion              |
| $b_0$        | 0.76554                | ${\rm kg^2m^{-3}g^{-1}}$                    | haline expansion               |
| $C_b$        | $9.9\times10^{-3}$     | $\rm kgm^{-3}K^{-2}$                        | thermal cabbeling              |
| $T_h$        | $2.47\times10^{-5}$    | $\rm kgm^{-4}K^{-1}$                        | thermobaric effect             |
| $A_{\Theta}$ | 40                     | ${\rm W}{\rm m}^{-2}{\rm K}^{-1}$           | $\Theta$ restoring coefficient |
| $A_S$        | $3.858 \times 10^{-3}$ | $\mathrm{kg}\mathrm{m}^{-2}\mathrm{s}^{-1}$ | $S_A$ restoring coefficient    |

# 2.2 Model domain

85

The DINO configuration is an idealized Atlantic sector model (Fig. 1). The domain is broadly adapted from the NW2 configuration. It spans  $50^{\circ}$  in longitude and approximately  $70^{\circ}$  north and south of the equator in latitude. The boundaries are closed everywhere except for a zonally periodic reentrant channel between  $45^{\circ}$  S and  $65^{\circ}$  S.

The bathymetry of DINO is a two-hemisphere extension of the configuration by Caneill et al. (2022), with a smooth exponential slope at boundaries and channel walls. The slope steepens from a maximum depth of 4000 m toward a minimum depth of 2000 m, above which the domain is bounded by vertical walls. A semicircular ridge at the western edge of the channel represents an idealized Scottia ridge. It introduces a horizontal pressure gradient to the channel at depth, which reduces momentum of the circumpolar flow and retroflects the deep western boundary currents. We do not add a Mid-Atlantic Ridge to the bathymetry, unlike in NW2. We found that this prohibits a coherent subpolar gyre and introduces an undesirable separation into two basins with respect to dense water formation and meridional overturning. While such a separation is indeed observed for the recirculation at depth, the idealized geometry and buoyancy forcing of DINO cannot capture this appropriately. The complete analytical formulation of the bathymetry can be found in Appendix A. For the momentum equations, we apply free-slip boundary conditions at the coast and a non-linear friction term at the bottom. The surface boundary conditions are described in the following.

## 2.3 Surface forcing

The surface boundary conditions for the prognostic equations are analytically prescribed by zonally uniform profiles (Fig. 2). Regarding momentum, we follow Marques et al. (2022) using a purely zonal wind stress profile  $\tau_u$  constructed by piecewise cubic interpolation between fixed values of 0, 0.2, -0.1, -0.02, -0.1, 0.1 and  $0 \,\mathrm{N}\,\mathrm{m}^{-2}$  for the latitudes -70, -45, -15, 0, 15, 45 and 70 ° N (panel (a) of Fig. 2). The wind stress applies to the momentum equations as a Neumann boundary condition of vertical momentum diffusion, but can also be interpreted as a constant source term to the topmost fluid layer of thickness

**Figure 1.** Bathymetry of the DINO configuration. The outlines of the continents are displayed in the background for reference only, in order to provide a comparison to the actual Atlantic Ocean. Lateral boundaries are closed everywhere except a periodic reentrant channel between  $45 \,^{\circ}$  S and  $65 \,^{\circ}$  S.

 $\Delta z_0$ .

$$\boldsymbol{F^u} = \frac{\tau_u}{\rho_0 \, \Delta z_0} \boldsymbol{i} \tag{7}$$

For the tracer equations, we apply Haney-type boundary conditions (Haney, 1971). Temperature and salinity are restored to meridional profiles  $\Theta^*$  and  $S^*$ , adapted from Munday et al. (2013) (panel (c) and (d) of Fig. 2). As for momentum, they enter the tracer equations as a source term to the topmost fluid layer.

$$F_{ns}^{\Theta} = \frac{1}{c_p \rho_0 \Delta z_0} \left( A_{\Theta} \left( \Theta^* - \Theta \right) - Q_{sr} \right) \tag{8}$$

and

$$F^{S_A} = \frac{A_S}{\rho_0 \Delta z_0} \left( S^* - S_A \right),\tag{9}$$

where we use the same temperature and salinity restoring coefficient  $A_{\Theta} = 40 \,\mathrm{W\,m^{-2}\,^{\circ}C^{-1}}$  and  $A_S = 3.858 \times 10^3 \,\mathrm{kg\,m^{-2}\,s^{-1}}$  as Caneill et al. (2022) and the specific heat capacity  $c_p = 3991.86 \,\mathrm{J\,kg^{-1}\,K^{-1}}$ . Note that Eq. 8 only represents the non-solar heat flux, where we subtract the solar heat flux from short wave radiation  $Q_{sr}$ . This radiative flux does not only enter the topmost fluid layer, but can penetrate the water column. Its absorption corresponds to oligotrophic type I water in the classification of optical properties of seawater by Jerlov (1968), with two wavebands of e-folding scale  $\zeta_0 = 0.35 \,\mathrm{m}$  and  $\zeta_1 = 23 \,\mathrm{m}$ .

Figure 2. Meridional profiles of the surface forcing fields: zonal wind stress (a), solar heat flux (b), temperature (c), salinity (d) and effective density (e) restoring. The density restoring is diagnosed from  $\Theta^*$  and  $S^*$  through the equation of state (Eq. 6). All profiles are uniform along the zonal direction and depict the yearly average. Blue shading indicates the minimum and maximum value due to the seasonal cycle.

115 The resulting source term in Eq. 2 is given by

$$F_{sr}^{\Theta} = \frac{Q_{sr}}{c_p \rho_0} \partial_z \left[ 0.58 e^{-\frac{z}{\zeta_0}} + 0.42 e^{-\frac{z}{\zeta_1}} \right]. \tag{10}$$

The solar heat flux follows a seasonal cycle (indicated by the blue shading in Fig. 2). The seasonal cycle of the temperature restoring profile lags by one month, mimicking an atmospheric response to insolation. Its amplitude is larger in the Northern Hemisphere. From  $\Theta^*$  and  $S^*$  we can diagnose an *effective density restoring*  $\rho^*$  (panel (e) in Fig. 2). Three key observations in  $\rho^*$  motivate our choice of restoring profiles:

- 1. They ensure that water forming at the southern boundary is always denser than the water forming at the northern boundary. This represents the model analog of North Atlantic Deep Water (NADW) located above denser Antarctic Bottom Water (AABW).
- The meridional density gradient is positive and approximately uniform throughout most of the periodic channel. This
   sets the strength of the ACC through thermal wind balance.
  - 3. Water forming at the northern boundary shares isopycnals outcropping within the periodic channel. This condition was found necessary to support a pole-to-pole component of the MOC (Wolfe and Cessi, 2011).

Complete analytical formulations of  $\Theta^{\star}$ ,  $S^{\star}$  and  $Q_{sr}$  are provided in Appendix B.

## 2.4 Diagnosing subgrid fluxes through coarse-graining

Since we aim to compare solutions of the described configuration across different horizontal resolutions, it is useful to introduce a coarse-graining formalism that allows us to average high-resolution results onto a lower resolution grid. We follow the approach of Mana and Zanna (2014) and use this method to diagnose eddy fluxes from the subfilter scale of the coarse-grained model solution, reflecting the unresolved subgrid scale of coarse model simulations. The idea is that a low-resolution model with ideal parameterizations of these unresolved processes should behave as the coarse-grained high-resolution model. For simplicity, we demonstrate the concept for the momentum equation (Eq. 1), but it also holds for the tracer equations.

The coarse-graining operator, denoted by an overbar (...), is a Reynolds average weighted by the local grid size. We assume that it only acts in the horizontal direction and commutes with spatial derivatives. While this assumption is not strictly valid for nonlinear dynamics discretized on a sphere (Aluie, 2019), we consider it sufficient for the purposes of this study. Applying the operator to Eq. 1 then yields

140 
$$\partial_t \overline{u}_h + \overline{\left[ (\nabla \times u) \times u + \frac{1}{2} \nabla u^2 \right]_h} + f \mathbf{k} \times \overline{u}_h + \frac{1}{\rho_0} \overline{\nabla}_h \overline{p} = \mathbf{D}^{\overline{u}} + \mathbf{F}^{\overline{u}}$$
 (11)

By adding the non-linear advection term of the coarse-grained velocity fields to both sides and rearranging the terms, we arrive at the coarse-grained momentum equation.

$$\partial_t \overline{\boldsymbol{u}}_h + \left[ \left( \overline{\boldsymbol{\nabla}} \times \overline{\boldsymbol{u}} \right) \times \overline{\boldsymbol{u}} + \frac{1}{2} \overline{\boldsymbol{\nabla}} \overline{\boldsymbol{u}}^2 \right]_h + f \, \boldsymbol{k} \times \overline{\boldsymbol{u}}_h + \frac{1}{\rho_0} \, \overline{\boldsymbol{\nabla}}_h \overline{\boldsymbol{p}} = \boldsymbol{D}^{\overline{\boldsymbol{u}}} + \boldsymbol{F}^{\overline{\boldsymbol{u}}} + \boldsymbol{\mathcal{S}}_{sgs}^{\boldsymbol{u}}$$

$$\tag{12}$$

with

145 
$$\mathcal{S}_{sgs}^{u} = \left[ \left( \overline{\nabla} \times \overline{u} \right) \times \overline{u} + \frac{1}{2} \overline{\nabla} \overline{u}^{2} \right]_{b} - \overline{\left[ (\nabla \times u) \times u + \frac{1}{2} \nabla u^{2} \right]_{b}}.$$
 (13)

Eq. 12 resembles Eq. 1 for the coarse velocity fields plus an additional eddy source term  $S_{sgs}^u$ . It arises from the non-linearity of the advection term and represents interactions of the resolved flow with the unresolved spatial scales (Aluie, 2019). From Eq. 12 we can derive the tendency for the kinetic energy budget due to the subgrid terms through the chain rule as follows.

$$\partial_t \left[ \frac{1}{2} \overline{u}^2 \right]_{sgs} = \overline{u} \cdot \left[ \partial_t \overline{u} \right]_{sgs} = u \cdot \mathcal{S}^u_{sgs} \tag{14}$$

Similar subgrid-scale interaction terms can be derived for the tracer equations as well, namely  $\mathcal{S}_{sgs}^{\Theta}$  and  $\mathcal{S}_{sgs}^{S}$ . Finding good approximations to these terms to close the equations is ultimately what developing parameterizations of subgrid-scale processes is all about.

#### 3 Numerical implementation in NEMO

In the previous section, we introduced the ingredients defining the DINO configuration. Now, we focus on some numerical choices, essential to arrive at the results presented here. These should be seen as user choices for the purposes of this study,

but can be adapted for other use cases. We use the well-established NEMO framework, version 4.2.1. For a more detailed description of its numerical methods, we refer to its documentation (Madec et al., 2023).

We use 36 vertical levels with a grid spacing increasing with depth, adapted from Lévy et al. (2010). The levels are time- and space-dependent as they are referenced to the sea surface height obtained from the non-linear free surface boundary (quasi-Eulerian, or  $z^*$  coordinates in Adcroft and Campin (2004)). In the horizontal direction, we chose an isotropic Mercator grid, so that the horizontal grid spacing  $\Delta x = \Delta y$  decreases poleward. This is convenient because the first baroclinic deformation radius  $L_D$  decreases with latitude similarly, and explicitly resolving mesoscale eddies requires that  $L_D$  is captured by at least two grid points (Hallberg, 2013).

We show results for DINO within three classes of models, categorized by their horizontal resolution following Hewitt et al. (2020): eddy-parameterizing with 1°, eddy-permitting with 1/4°, and eddy-resolving with 1/16° zonal grid spacing. The corresponding reference experiments are denoted R1, R4, and R16, respectively. A detailed description of the spatial discretization is provided in Appendix C.

For the Coriolis and advection term in the momentum equations, we chose an energy- and enstrophy-conserving advection scheme in vector invariant form. (see Chapter 5.2 of Madec et al. (2023)). In the tracer equations, the flux-corrected transport (FCT) scheme is employed.




The pressure gradient term is computed directly by a cumulative sum of horizontal density gradients from the surface to the bottom, where the surface pressure follows from the nonlinear free surface formulation of NEMO. The forcing terms are introduced in section 2.3 and formulated purely analytically. They are simply computed on the model grid and applied as source terms to the right-hand sides. The remaining terms in Eq. 1 to Eq. 3, denoted as  $D^{u/\Theta/S_A}$ , represent subgrid parameterizations and can vary depending on the specific experiment and the resolution of the model. We present DINO as a testbed for horizontal closures, so parameterizations for vertical mixing do not vary between experiments. We use the Turbulent Kinetic Energy (TKE) closure adapted from Blanke and Delecluse (1993) with a background vertical eddy viscosity of  $1.2 \times 10^{-4}$  m<sup>2</sup> s<sup>-1</sup> and diffusivity of  $1.2 \times 10^{-5}$  m<sup>2</sup> s<sup>-1</sup>. Where the model becomes statically unstable, the vertical eddy mixing coefficients are drastically increased to  $100 \text{ m}^2 \text{ s}^{-1}$  to mimic a fast convective adjustment.

For R1, momentum and tracers are dissipated by Laplacian friction along isopycnal surfaces. The diffusivity and viscosity parameters scale linearly with the grid spacing. For higher horizontal resolution (R4 and R16), we replace the Laplacian friction with a bilaplacian operator. The diffusivity parameter then scales cubically with the grid spacing, following Willebrand et al. (2001). For viscosity, we chose a grid- and flow-dependent model proposed by Smagorinsky (1963). Our choices are motivated by the need to maintain numerical stability while avoiding excessive dissipation of the eddy field. At eddy-parameterizing resolution, we employ the GM parameterization mentioned in section 1 to account for the unresolved tracer transport by mesoscale eddies. The GM coefficient is chosen to be dependent on the local growth rate of baroclinic instability, as suggested by Tréguier et al. (1997). In eddy-permitting and eddy-resolving horizontal resolution, we assume this process to be at least partially resolved and consequently omit the GM parameterization. A summary of the different parameterizations along with their numerical parameter values is given in Table 2.

**Table 2.** Differences between DINO reference experiments at eddy-parameterizing, eddy-permitting and eddy-resolving horizontal resolutions. The indicated hCPU are only estimates for 1 simulated year on the Jean Zay supercomputer used for this study and cannot be generalized.

|                                    | R1                                                                                      | R4                                                 | R16              |
|------------------------------------|-----------------------------------------------------------------------------------------|----------------------------------------------------|------------------|
| $\Delta x/\Delta y$ at the equator | 111.1 km                                                                                | $27.8\mathrm{km}$                                  | $6.9\mathrm{km}$ |
| $\Delta x/\Delta y$ at the poles   | $38.0\mathrm{km}$                                                                       | $9.5\mathrm{km}$                                   | $2.3\mathrm{km}$ |
| $\Delta t$                         | $45\mathrm{min}$                                                                        | 15 min                                             | $3 \min$         |
| hCPU / simulated year              | 2                                                                                       | 120                                                | 9000             |
| Lateral diffusivity                | $\nabla^2$ , $\kappa_T = \frac{1}{2} U_T \Delta x$                                      | $\nabla^4, \kappa_T = \frac{1}{12} U_T \Delta x^3$ | same as R4       |
|                                    | with $U_T = 0.027  \mathrm{m  s^{-1}}$                                                  | with $U_T = 7.68 \times 10^{-4} \mathrm{ms^{-1}}$  |                  |
| Lateral viscosity                  | $ abla^2,  \kappa_M = \frac{1}{2}  U_M  \Delta x$ with $U_M = 0.27  \mathrm{m  s^{-1}}$ | $ abla^4$ , Smagorinsky (1963) with $C_{smag}=3.5$ | same as R4       |
| Eddy induced advection             | Tréguier et al. (1997)                                                                  | explicit                                           | explicit         |

# 190 4 Spin-up and model experiments




DINO incorporates diabatic processes, such as dense water formation and diapycnal mixing. These are slow mechanisms that adjust the stratification to perturbations over centuries. Consequently, the spin-up of DINO takes a few thousand years to equilibrate. This is computationally too expensive for eddy-resolving horizontal resolution. Therefore, we compute the spin-up with 1° zonal grid spacing. We start from rest with idealized initial temperature and salinity fields. Their complete analytical description is provided in Appendix D. The model is integrated for 3000 years, until the tracer and velocity fields have reached a quasi-equilibrated state (see Appendix E for details on equilibration). We then use the spun-up state as new initial conditions for all other experiments (Fig. 3). For experiments with a higher horizontal resolution, the tracer fields are first interpolated onto the respective grid using the Python package *Xesmf* (Zhuang et al., 2023). The interpolation method we used was designed for scalar fields and cannot ensure conservation of some key properties of vector fields, such as divergence or vorticity. Since the velocity fields spin up rather quickly, we chose to initialize all experiments from rest, after interpolating only the tracer and sea surface height fields.

In the following sections, we present two intercomparisons of DINO at different resolutions. In Section 5, we compare R1 and R4 to illustrate parameterized and partially resolved eddy-induced tracer transport across the two resolution regimes. We analyze its effect on the mean circulation and stratification fields. Hence, we afford 400 years of integration time for both experiments to allow for the slow adjustment of the mean state to the change in resolution.

At  $1/16^{\circ}$  horizontal resolution we cannot afford this long adjustment. Consequently, we compare only 30 years of the R4 and R16 experiments in Section 6. This integration time was found to be sufficient for faster adjusting metrics, such as kinetic energy and its transfer across scales. Fortunately, it is precisely these fast adjusting processes in the energy cycle which are

Figure 3. Schematic representation of the experiment design. R1, R4 and R16 denote experiments at  $1^{\circ}$ ,  $1/4^{\circ}$  and  $1/16^{\circ}$  horizontal resolution, respectively. C16 is the R16 output, but coarse-grained to the same resolution as R4, following subsection 2.4. The thick black line denotes the initial state for all reference experiments, interpolated from the R1 spin-up to higher horizontal resolutions where necessary. Gray boxes indicate the last 50 years of the 400 year simulations and the last 10 years of the 30 year simulations used for data collection. They also mark the experiments used for the eddy-permitting/eddy-parameterizing and eddy-resolving/eddy-permitting comparison in section 5 and section 6.

addressed by parameterizations for the eddy-permitting regime. However, we cannot make quantitative statements about their effect on the mean state with the short experiments presented here.

Furthermore, we coarse-grain the output of the high-resolution data to the same resolution as R4, denoted as C16. As described in Section 2.4, we use an area-weighted Reynolds average on snapshots of the model output for coarse-graining. The convolution is computed at the cell center for all fields, so the velocity components are first interpolated to T-points. This approach treats the velocity components as scalar fields and does not strictly preserve the conservation properties of the 3D vector field. However, it is the simplest approach and considered sufficient for the purposes presented in this study. Before coarse-graining, we smoothen the velocity fields with a simple fixed-factor filter of the *GCM-filters* python package (Grooms et al., 2021; Loose et al., 2022), to suppress high-amplitude noise at the grid scale (Perezhogin et al., 2023b).

In the following, we climb the ocean equivalent of Charney's famous ladder (Balaji, 2021) in two steps: from the eddy-parameterizing into the eddy-permitting regime in Section 5 and further into the eddy-resolving regime in Section 6.

#### 220 5 The eddy-parameterizing regime



DINO simulates an idealized Atlantic Ocean circulation and reproduces essential features such as the ACC, MOC, subtropical and subpolar gyres, western boundary currents, and dense water formation. In this section, we compare the mean state circulation of DINO for the eddy parameterizing and eddy-permitting reference experiments R1 and R4, respectively.

The barotropic transport stream function reflects the volume transport of the vertically averaged horizontal flow field. Fig. 4 shows the time average over the last 50 years of data collection. The horizontal flow depicted in Fig. 4 is similar for both experiments: a system of alternating subpolar, subtropical, and tropical gyres and the intense circumpolar current at the zonally

**Figure 4.** Barotropic transport stream function for (a)  $1^{\circ}$  reference (R1), (b)  $1/4^{\circ}$  reference (R4). Time average over the last 50 years of simulation.

periodic reentrant channel. For R1, the flow is mostly laminar, while it is disturbed by partially resolved geostrophic turbulence for R4. The western flank of the subtropical gyres and its zonal extension along  $40^{\circ}$  N strengthen with increased horizontal resolution. In contrast, ACC transport is weaker for the R4 case.




Fig. 5 complements the barotropic transport in Fig. 4 with the vertical structure of the circulation. It shows the MOC transport stream function in potential density space, referenced to  $2000\,\mathrm{m}$ . For the lightest water masses above roughly  $32\,\mathrm{kg\,m^{-3}}$  we find two shallow tropical cells, antisymmetric about the equator. Just below and further poleward extends a second pair of overturning cells, visible between  $32\,\mathrm{kg\,m^{-3}}$  and  $34\,\mathrm{kg\,m^{-3}}$ . These cells are an imprint of the tropical and subtropical gyres on the meridional overturning, transporting light, warm waters poleward and colder, denser water equatorward. Below approximately  $34\,\mathrm{kg\,m^{-3}}$  the antisymmetry about the equator breaks. There is a distinct overturning cell associated with the subpolar gyre apparent in the Northern Hemisphere, but its return flow partially crosses the equator along  $36\,\mathrm{kg\,m^{-3}}$  iso-lines as part of a pole-to-pole overturning circulation. A deep anti-clockwise overturning cell develops for the densest water masses below  $36\,\mathrm{kg\,m^{-3}}$  in R4, but not in R1. We can associate the overturning to model analogues of water masses by having a look at zonally averaged potential density contours, referenced to  $2000\,\mathrm{m}$  (Fig. 6). The deep cell just below the iso-lines of  $36\,\mathrm{kg\,m^{-3}}$  transports the model analogue of Antarctic Bottom Water (AABW), formed by convection close to the southern boundary. The flow of the upper cell is along isopycnals, which outcrop at the northern boundary and the southern end of the periodic channel. This allows interhemispheric, adiabatic transport of North Atlantic Deep Water (NADW) formed by convection at the northern boundary. The circulation and associated stratification are broadly similar for both horizontal resolutions and fit the theoretical

**Figure 5.** Meridional Overturning Circulation. Contours of the zonally averaged transport streamfunction in density space for the (a) R1 and (b) R4 reference experiment. Solid contours with red shading represent clockwise, dashed contours with blue shading anti-clockwise circulation. The potential density anomalies are referenced to 2000 m depth. Time average over the last 50 years of simulation.

framework discussed in previous studies (Vallis, 2017; Wolfe and Cessi, 2011; Munday et al., 2013). However, the transport of the diabatic cell in Fig. 5 is larger for R1 compared to R4. And the deep cell in the Southern Ocean almost vanishes for the R1 experiment. This occurs alongside a volume reduction of AABW and a volume increase of NADW, with steeper isopycnals in the Southern Ocean channel.




The AMOC plays an essential role in regulating Earth's climate system, primarily by transporting heat from the tropics to the North Atlantic. This is illustrated in Fig. 7, which shows the zonally and vertically integrated meridional heat transport for the R1 and R4 experiments. The mean transport (solid lines) for R1 is oriented northward everywhere, except for a small band close to the southern boundary where the transport is southward. It is also overall more northward than for the R4 experiment, where we find wider bands of southward transport in the Southern Hemisphere. This partially reflects the changes in circulation described above. A weaker, cross-equatorial meridional overturning in the R4 experiment leads to less northward heat transport. Besides, a strengthened circulation of AABW leads to more southward heat transport in the Southern Hemisphere. We separate the eddy component of meridional heat transport, indicated by the dashed lines in Fig. 7. It is computed as the residual of an average taken over the last 50 years of data. For R1, the eddy induced advection of heat is almost entirely due to the GM parameterization. The eddy-induced heat transport is remarkably similar for the partially resolved eddy field in R4 and the parameterized one in R1. The most significant difference occurs in the tropics, where large eddies can be explicitly resolved

Figure 6. Zonally averaged contours of potential density referenced to  $2000 \,\mathrm{m}$  depth ( $\sigma_2$ ) for the (a) R1 and (b) R4 experiment. Dashed vertical lines mark the channel position. Time average over the last 50 years of simulation.

**Figure 7.** Meridional heat transport as a function of latitude in R1 and R4, produced by the mean currents (straight lines) and by mainly the parameterized (R1) or partially resolved (R4) eddies (dashed lines). Time average over the last 50 years of simulation.

at 1/4° resolution and partially resolved at 1° resolution. Here, GM is deactivated, causing a very small eddy induced heat transport. It follows that the striking differences in meridional heat transport between R1 and R4 are not a direct consequence of the partially resolved or parameterized eddy transport, but rather an indirect result of changes in the mean circulation.

To better understand this process we have a closer look at the role of eddies in the Southern Ocean. The strong eastward winds, combined with the absence of zonal boundaries, drive the Antarctic Circumpolar Current (Fig. 4). In thermal wind balance, this intense current is characterized by steeply sloping isopycnals. At  $1/4^{\circ}$  horizontal resolution, we start to resolve large eddies. In their formation through baroclinic instabilities, they extract APE and thereby act to flatten these steep isopycnals, while decelerating the ACC. We cannot explicitly resolve this physical process at  $1^{\circ}$  horizontal resolution in high latitudes. Instead, we parameterize it with the GM-scheme. Fig. 6 suggests that GM with the parameter choice presented here is less efficient at extracting APE than the partially resolved eddy field of R4, leading to steeper isopycnals and a stronger ACC. In the R1 experiment, we observe an ACC transport of  $206.0 \, \text{Sv}$ , whereas it is reduced to  $149.7 \, \text{Sv}$  in the R4 experiment (see Fig. 4). It is through this delicate balance that the parameterized eddies of R1 or the partially resolved eddies of R4 are setting the global stratification and hence circulation.

The presented results show that the idealized DINO configuration is capable of simulating the primary mechanisms through which horizontal resolution influences the mean circulation. We illustrate its utility in testing the impact of eddy parameterizations on the mean state, using the GM parameterization as a case study. It should be noted that the GM coefficient could be tuned to better match the R4 experiments regarding the shown metrics. We chose to use standard values of a commonly used scheme in NEMO to demonstrate the test protocol with a widely recognized subgrid parameterization.

At eddy-permitting horizontal resolution, only the largest scales of geostrophic turbulence ( $\sim 50$  to  $200\,\mathrm{km}$ ) are explicitly resolved, but not its entire spectrum. The unresolved small scales have a major impact on the energy cycle. In the following, we proceed to eddy-resolving horizontal resolution to showcase the processes still missed in the eddy-permitting regime.

#### 6 The eddy-permitting regime







Fig. 8 shows a time series of kinetic energy integrated over the whole domain for the reference experiments R4, R16, and the coarse-grained high-resolution experiment C16 for the 30 years integration period shown in Fig. 3. We use the last 10 years of simulation for data collection (indicated by the blue shading in Fig. 8). The kinetic energy exhibits a slow drift resulting from changes in the mean circulation, discussed in the previous section. However, it occurs on much slower timescales than the processes studied in this section and is considered negligible for our purposes. The total kinetic energy in the R16 experiment is more than two and a half times higher compared to the R4 experiment, namely  $2.1 \times 10^{18}$  J as compared to  $0.8 \times 10^{18}$  J, respectively. This drastic increase with horizontal resolution is only partially captured by the added range of smaller scales resolved in R16. After removing them by coarse-graining to the same resolution as R4 (C16), most of the total kinetic energy is retained. This suggests that a large portion of KE introduced at smaller scales is transferred to larger scales, which are resolved by R4. The process behind this phenomenon is an upscale energy transfer known as the *inverse energy cascade* in turbulence theory. In the eddy-permitting regime, this source of kinetic energy is missing and needs to be parameterized.

With the coarse-graining approach applied to the high-resolution experiment, we can directly diagnose the subgrid eddy momentum fluxes, following Eq. 13. Fig. 9 shows surface KE snapshots for R4, R16, and C16 (panel (a) to (c)), as well as the instantaneous KE tendency due to the subgrid forcing, as derived in Eq. 14 (panel (d)). The surface flow is more energetic for

**Figure 8.** Total kinetic energy for R4 (pink), R16 (black) and C16 (dashed black). The light-blue shading indicates the time span used for data-collection.

**Figure 9.** Surface kinetic energy snapshots for the (a) R4, (b) R16 and (c) C16 reference experiment. Panel (d) shows the KE tendency per unit mass due to the subfilter eddy source term of the C16 experiment. White boxes indicate the Gulf-Stream (top) and Southern Ocean (bottom) region. All snapshots are computed for the last time step after 30 years of simulation.

the R16 experiment throughout the domain. A fine mesh of small-scale turbulence is revealed at higher resolution, which is not resolved in the R4 experiment. Additionally, large-scale filaments show a notable increase of KE, mainly in the Gulf Stream

region (upper white box), Southern Ocean region (lower white box), and around the Equator. A large part of the increase is retained after coarse-graining. Again, this demonstrates the inverse energy cascade that transfers KE from smaller to larger scales. It is instantaneously captured by the subgrid forcing of KE in panel (d). However, the simultaneous injection and extraction of KE at various scales obscures any evidence for such an upscale transfer. To better quantify the phenomenon, we analyze the KE spectra across two distinct dynamical regimes: the Gulf Stream region and the Southern Ocean (see Fig. 10). We compute the spectra with a 2D Fourier transform of the instantaneous velocity fields, interpolated to an equidistant Cartesian grid. We use the Python library xrft (Zhuang et al., 2023) and apply a linear detrending along with a Hann window. By integrating along circles of constant wave numbers in the spectral x-y plane, we obtain the isotropic power density spectra.


Figure 10. The upper panels depict the kinetic energy spectrum for the R4, R16, and C16 experiments in the (a) Southern Ocean and (b) Gulf Stream regions. The gray dashed line represents a  $k^{-5/3}$  power law. The lower panels, (c) and (d), show the corresponding kinetic energy transfer from the sub-filter scale, computed as a cross-spectrum between the sub-grid forcing  $S_{sgs}$  and the coarse-grained velocities. All spectra are derived from surface velocity fields within the white boxes in Fig. 9, interpolated onto an equidistant Cartesian grid.

Panel (a) of Fig. 10 shows power density spectra of surface KE in the Southern Ocean region, indicated by the lower white rectangle in Fig. 9. We include a logarithmic slope  $k^{-\frac{5}{3}}$ , which is predicted for the inverse energy cascade by theory (Graham and Ringler, 2013). The R16 spectrum (in black) follows the slope of  $k^{-\frac{5}{3}}$  from length scales of approximately  $40 \, \mathrm{km}$  to  $200 \, \mathrm{km}$ , where it has its maximum. Below length scales of  $40 \, \mathrm{km}$  it steepens rapidly towards the grid scale where kinetic energy is dissipated by numerical and explicit diffusion. For coarser horizontal resolution (R4), the KE is lower across all scales, but has a broadly similar shape to that of R16. The spectrum of C16 resembles that of R4 close to the grid-/filter scale.

Towards larger scales, it approaches R16, as these scales remain unaffected by the coarse-graining operator. Panel (c) shows the KE transfer due to the sub-grid flux  $\mathcal{S}^{u}_{sgs}$  in C16 as given by Eq. 14. On average, the sub-grid fluxes extract KE for length scales smaller than 70 km and inject it at larger scales. This agrees well with the range of scales predicted by the  $k^{-\frac{5}{3}}$  power law fitted to the R16 spectrum. The KE transfer has a maximum of  $3.0 \times 10^{-9}$  m<sup>2</sup> s<sup>-3</sup> injected at scales just above 200 km and a minimum of  $-1.2 \times 10^{-9}$  m<sup>2</sup> s<sup>-3</sup> extracted at scales around 40 km. When we integrate the mean KE tendency per unit mass across the entire region, we find that the subgrid flux leads to a net increase of KE by  $9.6 \times 10^{-12}$  m<sup>2</sup> s<sup>-3</sup>.

For the Gulf Stream region (see panel (b)) all power density spectra are considerably lower in KE than those for the Southern Ocean region, across all scales. They are shifted slightly towards larger length scales, due to the increased grid spacing for lower latitudes. The same holds true for the KE transfer shown in panel (d). It is about one order of magnitude smaller in the Gulf Stream region. Here, KE is injected above and extracted below horizontal scales of 80 km. The maxima and minima of the transfer are found at similar horizontal scales as in the Southern Ocean, namely just above 200 km and around 50 km with KE transfers of  $2.0 \times 10^{-10}$  m<sup>2</sup> s<sup>-3</sup> and  $-0.4 \times 10^{-10}$  m<sup>2</sup> s<sup>-3</sup>, respectively. The net KE tendency per unit mass averaged across the Gulf Stream region is 1.8e - 12 m<sup>2</sup> s<sup>-3</sup>. As for the Southern Ocean, this represents a net increase of total KE in the Gulf Stream region. Once again, this illustrates the cascade of kinetic energy from the subfilter scale to the resolved flow at eddy-permitting resolution, explaining the weakened total KE of R4, where these scales remain unresolved (see Fig. 8).

#### 7 Discussion





DINO is an idealized model configuration, and as such, comes with inherent limitations. In this section, we discuss some of these limitations and point to possible future improvements.

The model is ocean-only and does not account for any feedbacks from the atmosphere, biosphere, or cryosphere. The latter two are entirely absent, while the atmospheric boundary is only represented through simplified, zonally uniform profiles. DINO also has an idealized bathymetry with no continental shelves, no mid-Atlantic ridge, a closed northern boundary, and an overall reduced volume when compared to the Atlantic. All these choices make it difficult to compare DINO output directly to observations. For example, the strength of the ACC can be modified by making small adjustments to the meridional surface density gradient through the temperature and salinity restoring (not shown). The meridional overturning and associated heat transport in DINO could only align with those of the real Atlantic by coincidence, as the volume and thermodynamic properties of the transported water masses are not the same. However, the lack of realism comes with the advantage of isolating the main processes influenced by horizontal resolution from the overwhelming complexity of Earth's climate system. It also enables simulations at high horizontal resolution for comparatively low computational cost.

Despite the low cost compared to global ocean models, we still encounter limitations regarding the horizontal resolution for this study. We afford 30 years of simulation for the R16 experiment, clearly not enough time for the mean state to fully adjust to the interpolation to higher resolution. Hence, we have no ground truth of the mean state of DINO. Nevertheless, we demonstrate in Section 6 that 30 years are adequate to study the subgrid momentum fluxes missed at eddy-permitting resolution. Moreover, the net effect of potential subgrid parameterizations on the mean state can still be evaluated, as demonstrated in Section 5.

For R16 we reach a horizontal grid spacing of a few kilometers in the polar oceans of DINO. While this is sufficient for the purposes of this study, we admit that the simulated outputs could possibly differ if spatial resolution were further increased. We could not afford to resolve the submesoscale regime in DINO, which plays a key role in the vertical restratification of the upper ocean and the forward kinetic energy cascade towards dissipation (Capet et al., 2008). We did not include any submesoscale parameterization, to avoid interference with the mesoscale parameterizations we are testing with DINO. That being said, it would be straightforward to extend the experiment design proposed in Section 4 with even higher resolution experiments when the computational resources are available.

In a step beyond the assessment of mesoscale parameterizations, DINO could be used to train machine-learning-based parameterizations as was done previously for simpler, more idealized models (Zanna and Bolton, 2020; Guillaumin and Zanna, 2021; Perezhogin et al., 2023a). The high-resolution data of R16 captures information on resolved geostrophic turbulence in the Southern Ocean, Gulf Stream extension, boundary currents, equatorial currents, and deep currents along topography. This richness in different dynamic regimes is an advantage over the previously mentioned studies.

# 8 Conclusions








We introduce the DINO configuration, a diabatic Atlantic basin model of intermediate complexity, designed for the testing of parameterizations in the eddy-parameterizing and eddy-permitting regime. For this purpose, we present results of DINO at horizontal resolutions of  $1^{\circ}$  (R1),  $1/4^{\circ}$  (R4) and  $1/16^{\circ}$  (R16).

We compare the mean state of R1 and R4 based on key metrics relevant to the climate system. We use the Gent and Mcwilliams (1990) parameterization as an illustrative example in our test configuration and showcase that, as implemented, it does not sufficiently extract available potential energy. Particularly in the Southern Ocean channel, this results in steeper isopycnals, subsequently accelerating the Antarctic Circumpolar Current through thermal wind balance from 149.7 Sv to 206.0 Sv. It also inhibits the formation of Antarctic Bottom Water and promotes the formation of North Atlantic Deep Water. Both these effects are reflected by the Meridional Overturning Circulation. Compared to R4, the circulation of R1 shows a weaker deep overturning cell associated with the transport of Antarctic Bottom Water and a stronger diabatic cell associated with the cross-equatorial transport of North Atlantic Deep Water. This also has consequences for the mean meridional heat transport, which we find to be overall more northward in the R1 experiment. This does not necessarily indicate a flaw in the Gent and Mcwilliams (1990) parameterization, as improved tuning could yield better results. Rather, it demonstrates that the DINO configuration captures useful metrics for the evaluation, development, tuning or training of novel parameterizations at coarse resolution.

Additionally, we compare instantaneous fields of the R4 and R16 experiment to quantify the kinetic energy transfers not resolved in the eddy-permitting regime. DINO's total kinetic energy at eddy-resolving resolution is about two and a half times higher than at eddy-permitting resolution. Using a simple coarse-graining approach, we provide evidence that most of the increase of kinetic energy in R16 is retained at scales resolved by R4, which we attribute to an inverse cascade of kinetic energy. We diagnose the missing subgrid fluxes of kinetic energy at eddy-permitting resolution from the coarse-grained high-

resolution data. On average, these fluxes extract kinetic energy below and inject kinetic energy above scales of around  $70 \,\mathrm{km}$  in the Southern Ocean and  $80 \,\mathrm{km}$  in the Gulf Stream region. The upscale transfer is almost one order of magnitude larger in the Southern Ocean. As for the eddy-parameterizing resolution, these results serve as benchmark metrics for developing and testing novel parameterizations in the eddy-permitting regime. In particular, they hold significant value for an emerging class of data-driven, machine-learning-based parameterizations.

Code and data availability. The source code and input files required to reproduce the DINO reference experiments presented in this study are available at https://doi.org/10.5281/zenodo.15016824 (Kamm, 2025a). The data and code to reproduce all figures are available at https://doi.org/10.5281/zenodo.15808281 (Kamm, 2025b).

Acknowledgements. We would like to thank the members of M<sup>2</sup>LInES, especially Laure Zanna, Alistair Adcroft, and Pavel Perezhogin for insightful discussions. We also thank Romain Caneill and Fabien Roquet for helping to set up the geometry, thermodynamics, and forcings of the proposed configuration. This work was granted access to the HPC resources of IDRIS under the allocation 2023-A0140107451 and 2024-A0160107451 made by GENCI. Support was provided by Schmidt Sciences, LLC.

Author contributions. DK, JD and GM shaped the experimental design of the study, DK and GM contributed to the implementation of DINO into NEMO, DK, JD led the manuscript and GM contributed to the manuscript.

Competing interests. The authors declare that they have no conflict of interest.


## Appendix A: Bathymetry



The shape of the bathymetry is defined through a normalized and tapered exponential function.

$$g(x, x_1, x_2, s, d) = \begin{cases} 1 - \frac{e^{-s(x-x_1)}}{1 + e^{-s\Delta\lambda}} (1 - S(x, x_1, x_1 + d)), & x_1 \le x \le x_1 + d \\ 1, & x_1 + d \le x \le x_2 - d \\ 1 - \frac{e^{s(x-x_2)}}{1 + e^{-s\Delta\lambda}} S(x, x_2 - d, x_2), & x_2 - d \le x \le x_2, \end{cases}$$
(A1)

where d is the tapering distance, s is a parameter controlling the slope,  $\Delta \lambda$  is the longitudinal width of the basin, x is one of the horizontal coordinates  $(\lambda, \varphi)$  with its respective boundary values  $x_1 < x_2$  and S is the smooth step function for tapering:

$$S(x,a,b) = \begin{cases} 0, & x < a \\ 6\left(\frac{x-a}{b-a}\right)^5 - 15\left(\frac{x-a}{b-a}\right)^4 + 10\left(\frac{x-a}{b-a}\right)^3, & a \le x \le b \\ 1 & x > b \end{cases}$$
 (A2)

The tapering ensures that the slope reaches the sea floor without discontinuities after a maximum distance of d from the boundary. This is particularly important to avoid discontinuities in the periodic channel, where opposing lateral boundaries are closest. Consequently, we chose half the channel width  $\Delta\varphi_c$  as a tapering distance and arrive at the bathymetry b as

$$b(\lambda,\varphi) = \underbrace{g\left(\varphi,\varphi_1,\varphi_2,s_{\varphi},\frac{\Delta\varphi_c}{2}\right)}_{g_{\varphi}} \cdot \underbrace{g\left(\lambda,\lambda_1,\lambda_2,s_{\lambda},\frac{\Delta\varphi_c}{2}\right)}_{g_{\lambda}} (H_{\max} - H_{\min}) + H_{\min}. \tag{A3}$$

where  $\varphi_1 \approx -70^\circ$  N,  $\varphi_2 \approx 70^\circ$  N,  $\lambda_1 = -50^\circ$  E,  $\lambda_1 = 0^\circ$  E define the domain extent and  $H_{max} = 2000$  m,  $H_{min} = 4000$  m the minimum and maximum depth of the bathymetry. A minimum depth of 2000 m is arguably too deep for an accurate representation of the continental bathymetry. Deep western boundary currents flow above 2000 m and their interactions with the continental slope impact the separation of the Gulf Stream (Zhang and Vallis, 2007). This mechanism is not captured here. We chose 2000 m to enable a hybrid vertical coordinate, which employs z-levels in the upper 1000 m and terrain-following  $\sigma$ -levels below. Z-levels near the surface are necessary to prevent large errors in the pressure gradient term, particularly near the equator, where it is not balanced by the Coriolis effect. Ultimately, we want DINO to be usable by ocean model developers in any vertical coordinate. Therefore, we did not revisit the bathymetry for this study, where only pure z-levels are employed. The slope parameter is chosen as  $s_\lambda = \frac{1}{3^\circ}$  and  $s_\varphi = \cos\left(\frac{\pi \varphi_{\max}}{180}\right) s_\lambda$  to account for the grid deformation of the Mercator projection. The channel width is chosen as  $\Delta \varphi_c = 20^\circ$ , ranging from  $\varphi_{c_1} = -65^\circ$ N to  $\varphi_{c_1} = -45^\circ$ N. It is added to the bathymetry when periodic boundary conditions are chosen, by modifying  $g_\lambda$  as

$$\tilde{g}_{\lambda} = g\left(\lambda, \lambda_{1}, \lambda_{2}, s_{\lambda}, \frac{\Delta\varphi_{c}}{2}\right) \cdot \left[1 - g\left(\varphi, \varphi_{c_{1}}, \varphi_{c_{1}}, s_{\lambda}, \frac{\Delta\varphi_{c}}{2}\right)\right] + g\left(\varphi, \varphi_{c_{1}}, \varphi_{c_{1}}, s_{\lambda}, \frac{\Delta\varphi_{c}}{2}\right)$$
(A4)

Finally, the semi-circular sill is added in the form of a Gaussian ring centered around the midpoint of the Drake passage  $(\lambda_m, \varphi_m) = (-50^{\circ} \,\mathrm{E}, -55^{\circ} \,\mathrm{N})$ , yielding the modified bathymetry as

$$\tilde{b}(\lambda,\varphi) = \begin{cases} b(\lambda,\varphi) + (H_{sill} - b(\lambda,\varphi)) \exp\left(\frac{-\left(\sqrt{(\lambda - \lambda_m)^2 + (\varphi - \varphi_m)^2} - \frac{\varphi_c}{2}\right)^2}{s^2}\right) S(\lambda,\lambda_m,\lambda_m + s), & \text{if } b(\lambda,\varphi) \le H_{sill} \\ b(\lambda,\varphi), & \text{otherwise,} \end{cases}$$
(A5)

where the depth of the sill is chosen as  $H_{sill} = 2500 \,\mathrm{m}$ .

# Appendix B: Surface forcing

The temperature and salinity restoring profiles are given by

$$\Theta^*(t,\varphi) = \Theta^*_{n/s}(t) + \left(\Theta^*_{eq} - \Theta^*_{n/s}(t)\right) \cos(\frac{\pi \varphi}{L_{\varphi}})$$
 (B1)

$$S^*(\varphi) = S_{n/s}^* + \left(S_{eq}^* - S_{n/s}^*\right) \left(1 + \cos(\frac{2\pi\varphi}{L_{\varphi}})\right) / 2 - 1.25 e^{-\varphi^2/7.5^2},\tag{B2}$$

where the subscript  $(...)_{n/s}$  denotes the restoration value at the northern or southern boundary, depending on whether  $\varphi$  is located in the Northern or Southern Hemisphere, respectively.  $L_{\varphi} = 140^{\circ}$  is the approximate meridional extend of the domain in degrees. The time dependency of  $\Theta^{\star}$  reflects an atmospheric response to the seasonal cycle of solar insolation expressed by

$$\Theta_n^* = 5 + 3\cos\left(\pi \frac{d(t) - 201}{180}\right)$$
 (B3)

$$\Theta_s^* = -0.5 - 0.5 \cos\left(\pi \frac{d(t) - 201}{180}\right) \tag{B4}$$

where d denotes the day of the year. For simplification, one year of DINO has 12 months with 30 days each. The seasonal cycle has its maximum and minimum on July and January  $21^{st}$ , lagging one month behind the solstices. It only modulates the meridional boundary values of the restoring temperature and leaves the equatorial value constant as  $\Theta_{eq}^* = 27.0\,^{\circ}\text{C}$ . In Eq. B2 the northernmost, southernmost, and equatorial restoring salinity are all constant and given by  $S_n^* = 35.0\,\text{g kg}^{-1}$ ,  $S_s^* = 35.1\,\text{g kg}^{-1}$ , and  $S_{eq}^* = 37.25\,\text{g kg}^{-1}$ , respectively.

The idealized meridional profile of short-wave radiative heat flux is given by

$$Q_{sr}(t,\varphi) = \max\left(230\cos\left(\frac{\pi}{180}\left[\varphi - 23.5\cos(\pi\frac{d(t) - 171}{180})\right]\right), 0\right). \tag{B5}$$

# Appendix C: Grid


Here, we provide details on the domain discretization. NEMO employs a staggered Arakawa C-grid (Arakawa and Lamb, 1977), and for clarity, all grid points mentioned in this section are defined at the *T*-point of that grid.

**Figure C1.** The ocean mesh of DINO at  $2^{\circ}$  horizontal resolution.

#### C1 Horizontal discretization

DINO is computed on a sphere and discretized by a horizontally isotropic Mercator grid (see Fig. C1). Therefore, the gridspacing in the latitudinal direction decreases towards the poles. We denote the grid indices in zonal and meridional directions as  $i \in [1, I]$  and  $j \in [1, J]$ , respectively. With a horizontal grid-spacing of  $\Delta \lambda$ , this defines the mesh as

$$\lambda(i) = \lambda_0 + \Delta\lambda * i \tag{C1}$$

$$\varphi(j) = \frac{180}{\pi} * \arcsin(\tanh(\Delta \lambda \frac{\pi}{180} * j))$$
 (C2)

I and J are chosen to span a domain of  $50^{\circ}$  from eastern to western boundary and approximately  $70^{\circ}$  from equator to both northern and southern boundary (not exactly as this cannot be guaranteed by Eq. C2). The longitudinal position of the domain  $\lambda_0$  is arbitrary, but chosen as  $50^{\circ}$  W to fit the Atlantic Ocean.

#### C2 Vertical discretization

In this study we chose K = 36 full step, quasi-Eulerian vertical levels. The depth at the vertical level midpoints (see Fig. C2)  $k \in [1.5, K + 0.5]$  is given by

$$z = a_2 + a_1 k + a_0 a_{cr} \ln \left( \cosh \left( \frac{k - k_{th}}{a_{cr}} \right) \right), \tag{C3}$$

where


$$\begin{split} a_0 &= \left(\Delta z_{min} - \frac{H}{K-1}\right) / \left(\tanh\left(\frac{1-k_{th}}{a_{cr}}\right) - a_{cr} \cdot \frac{\ln\left(\cosh\left(\frac{K-k_{th}}{a_{cr}}\right)\right) - \ln\left(\cosh\left(\frac{1-k_{th}}{a_{cr}}\right)\right)}{K-1}\right) \\ a_1 &= \Delta z_{min} - a_0 \cdot \tanh\left(\frac{1-k_{th}}{a_{cr}}\right) \\ a_2 &= -a_1 - a_0 \cdot a_{cr} \cdot \ln\left(\cosh\left(\frac{1-k_{th}}{a_{cr}}\right)\right), \end{split}$$

and H = 4000 m is the bottom depth,  $\Delta z_{min} = 10$  m the minimum level thickness,  $k_{th} = 35$  the index of the inflection point, and  $a_{cr} = 10.5$  the slope of the *tanh*.

**Figure C2.** Adapted from Figure 5 of Madec and Imbard (1996). The solid black line shows the depth for each vertical level, corresponding to the left y-axis. The dashed line shows the respective layer thickness indicated by the right y-axis.

## **Appendix D: Initial Conditions**

The ocean is initialized at rest for all experiments described in this work.

$$u(\lambda, \varphi, z) = 0, \quad v(\lambda, \varphi, z) = 0.$$
 (D1)

The vertical temperature and salinity profiles are defined through a combination of hyperbolic tangent functions, taken from the GYRE configuration (Lévy et al., 2010).

$$\Theta(z) = \left[16 - 12 \tanh\left(\frac{z - 400}{700}\right)\right] \frac{1 - \tanh\left(\frac{500 - z}{150}\right)}{2} + \left[15\left(1 - \tanh\left(\frac{z - 50}{1500}\right)\right) - 1.4 \tanh\left(\frac{z - 100}{100}\right) + 7\left(\frac{1500 - z}{1500}\right)\right] \frac{1 - \tanh\left(\frac{z - 500}{150}\right)}{2}$$

$$S_A(z) = \left[36.25 - 1.13 \tanh\left(\frac{z - 305}{460}\right)\right] \frac{1 - \tanh\left(\frac{500 - z}{150}\right)}{2} + \frac{1 - \tanh\left(\frac{z - 500}{150}\right)}{2} \dots$$
(D2)

We introduce a horizontal gradient from the equator to the poles with a simple linear transition of temperature and salinity towards bottom values of  $\Theta(z)$  and  $S_A(z)$  at the poles:

$$\tilde{\Theta}(\varphi, z) = \left( (\Theta(z) - \Theta|_{z=0}) \frac{\varphi_1 - |\varphi|}{\varphi_1} + \Theta|_{z=0} \right)$$
(D4)

$$\tilde{S}_A(\varphi, z) = \left( (S_A(z) - S_A|_{z=0}) \frac{\varphi_1 - |\varphi|}{\varphi_1} + S_A|_{z=0} \right)$$
(D5)

## **Appendix E: Spin-up strategy**

In Fig. 3 we present a schematic of the spin-up and experiment design employed throughout this study. Here we want to provide insights into how well equilibrated each experiment is with respect to the slowest adjusting processes, e.g. dense water formation. Hence, Fig. E1 shows the water mass proportions of the spin-up and each experiment with respect to our idealized definition of Antarctic Bottom Water ( $\sigma_2 > 36~{\rm kg}~{\rm m}^{-3}$ ), North Atlantic Deep Water ( $35~{\rm kg}~{\rm m}^{-3} \le \sigma_2 \le 36~{\rm kg}~{\rm m}^{-3}$ ) and surface water ( $\sigma_2 

Figure E1. Time series of the water masses partition based on an idealized definition of Antarctic Bottom Water ( $\sigma_2 > 36 \text{ kg m}^{-3}$ ), North Atlantic Deep Water ( $35 \text{ kg m}^{-3} \le \sigma_2 \le 36 \text{ kg m}^{-3}$ ) and surface water ( $\sigma_2 < 35 \text{ kg m}^{-3}$ ). Values are given as percentages of the total water volume. The 3000 years of spin-up at 1° horizontal resolution (panel (a)) is continued as experiment R1 for 400 years (panel (b)). R4 (panel (c)) and R16 (panel (e)) are initialized from the same stratification field as R1, indicated by the thick black line. The R4 experiment is continued for 400 years (panel (d)). Hatched areas indicate the period of data-collection for each experiment, corresponding to the gray boxes in Fig. 3.

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
