# Peer review of "DINO: A Diabatic Model of Pole-to-Pole Ocean Dynamics to Assess Subgrid Parameterizations across Horizontal Scales"

_EGUsphere, 2025_

## Author Comment (AC1)

**Reply to Referee comment #1:**

The authors would like to thank the referee for their time, as well as their invaluable comments and suggestions. In the following each comment, suggestion or concern is replied in purple font. Specific revisions of the text are in quotes, with the respective changes highlighted in bold.
* * *
**Summary and recommendation**

This manuscript proposes a modeling hierarchy, DINO, intended to act as a testbed for eddy parameterizations. The new hierarchy extends NeverWorld2, a previous such testbed, by including both temperature and salinity, an (idealized) nonlinear equation of state, an inter hemispheric overturning circulation, and diabatic processes. All of these either influence or are influenced by mesoscale eddies, so DINO would provide a more stringent and comprehensive test of eddy parameterizations. The ocean modeling community is in desperate need of such standardized testbeds and DINO stands to be a very useful contribution. The manuscript is generally well-written and the hierarchy carefully documented, with a few exceptions detailed under my specific comments. The comments primarily ask for clarification, although I do have concerns about the design of the freshwater forcing and the shortness of the period analyzed. I support publication if the authors can address these comments and concerns.

**Specific comments**

1. Two of the design decisions seem unusual or arbitrary. While they are unlikely to impact the ability of DINO to serve as a testbed for parameterizations, they deserve a few lines of additional justification
    1. The reentrant part of the domain spans 20°. This is significantly wider than the width of Drake Passage, which is about 8° wide. What is the rationale for choosing this width? To match NeverWorld2?

        In this we follow previous studies using sector models: NeverWorld2 (Marques et al 2022), Munday et al. 2013. Also, we note that the width of the southern ocean away from Drake passage is wider than 20° in latitude, so we do not see a need to revisit this choice.

    2. Why is a minimum depth 2000 m? Neverworld2 has a minimum depth of 200 m, which is a reasonable (if deep) value for continental shelves. In nature, the upper part of the North Atlantic deep western boundary current (associated with Labrador Sea Water) is found around 1000 m depth (Bower and Hunt 2000) and the interactions of the DWBC with the slope are thought to impact the Gulf Stream (Zhang and Vallis 2007). It would thus seem desirable to have the DWBC flow along the sloping topography rather than against the free-slip wall. However, Figure 5 shows that most of the southward flow of the mid-depth overturning cell is found at densities of 27 kg m$^{-3}$ or lighter, which figure 6 shows is shallower than 2000 m.

We agree with the comment. DINO was initially configured with a hybrid vertical coordinate, employing z-coordinates for the upper 1000 m and terrain-following coordinates below. We found that the terrain-following coordinates improve the Western Boundary Current structure and the Gulf Stream separation, while the z-coordinates near the surface prevent large errors in the pressure gradient term. The latter is especially important at the equator, where the pressure gradient is not balanced by the Coriolis effect. In order for the sigma-coordinates to smoothly transition to flat coordinates, we need the bathymetry to stay well below 1000 m, hence we chose 2000 m. As the parameterizations we are currently testing are developed for z-level coordinates only, the paper therefore shows results from DINO configuration with z-levels as well. Ultimately, we want DINO to be used by ocean model developers in any vertical coordinates, hence we did not revisit the bathymetry. It should also be noted that a shelf similar to NW2 would be difficult to represent in a coarse 1° model. We have inserted a short paragraph of justification for the bathymetry in Appendix A, summarizing the above.

2. It should be clarified that the equation of state (equation 6) is not an approximation to the *in situ* density, but the potential density (apparently referenced to the surface). The *in situ* density has a pressure dependence that leads to a nearly linear increase in density of about 4.5 kg m$^{-3}$ per km of depth. This, if the density at the surface is about 1026 kg m$^{-3}$, the density at 2000 m should be about 1035 kg m$^{-3}$. The potential density referenced to 2000 m (used in figures 5 and 6) should therefore be in the 30s rather than the 20s. It might be simpler to use potential density referenced to the surface in these figures—the numerical values are unlikely to change much, but they'd be closer to what people would expect for potential density.

This should indeed be clarified. The S-EOS approximates the in situ density minus a reference density profile. This is responsible for the mentioned differences to the potential density values at 2000m one would usually expect. The reason Roquet et al. (2015a and 2015b) could remove this background density profile is that it does not produce any dynamical effect. This is due to the fact that horizontal pressure gradients are dependent on horizontal density gradients only, which themselves are insensitive to the addition or removal of a vertical density profile in the equation of state. We added a clarification of the above where the EOS is first introduced.

3. Lines 112–114: Note that AABW and NADW have essentially the same density at the surface, but AABW is denser than NADW at depth due to the thermobaric effect (Nycander et al. 2015). Since DINO's equation of state supports the thermobaric effect, surface forcing that produces AABW that is denser than NADW at the surface may result in AABW that is excessively dense at depth.

Since DINO does not include sea ice and the idealized bathymetry has no shelf, the concept of AABW in DINO is only a very idealized model equivalent (as for NADW). But we agree with the reviewer and to clarify for the reader, we have rephrased the

mentioned lines: "***To ensure that*** water forming at the southern boundary is always denser than the water forming at the northern boundary…"

4. Lines 189–190: It is not clear how starting from rest ensures conservation or what is being conserved.

    We agree. We propose the following reformulation:

    "**The used interpolation tool only treats scalar fields and cannot ensure to conserve properties of vector fields, such as divergence, or vorticity. Since the velocity fields spin-up rather quickly in DINO, we chose to initialize all experiments from rest, after interpolating only the tracer fields and ssh.**"

5. The approach to freshwater forcing does not seem adequate. Salinity restoring is indeed unrealistic, but five years is unlikely to be sufficient to produce a stable climatology of moisture fluxes and four years is not long enough for the circulation to adjust to the change in the boundary conditions. Since the procedure for producing the freshwater forcing is repeated independently for each model resolution, this leads to each resolution being subjected to different freshwater forcing. This is undesirable for a model hierarchy that is supposed to only differ by resolution and subgrid scale parameterizations. In lieu of devising a new freshwater forcing scheme (which would require expensive recomputations), it would be more straightforward and clarifying to simply forgo freshwater forcing and analyze the cases with salinity restoring.

    The initial motivation behind this approach was to avoid damping the tracer variability by restoring to zonally uniform profiles of T and S. But given that in our approach, we force with time-mean EmP climatologies computed from S restoring while T remains forced through restoring, we agree that the advantage vanishes compared to the downsides of having different freshwater forcing across the hierarchy. We analyzed the results with salinity restoring only and it does not change the conclusions. We agree that this is a more straightforward forcing strategy, so we revised the manuscript to report only on simulations with salinity restoring and updated all figures accordingly.

6. Similarly, four years does not seem sufficient to characterize the mean state of higher resolution models.

    We agree with the reviewer that the highest resolution configuration did not run long enough to characterize the mean state, hence we only include fast-adjusting variables in our analysis (such as kinetic energy and associated spatial spectra). To make sure that our procedure reads clearly, the higher resolution model ran for respectively 23 years, and we averaged the final 4 years to produce mean variables that are finally analyzed.

    As we investigated for comments 5. + 6., we decided to extend our initial simulation of 19 years with salinity restoring up to 30 years. Preliminary tests showed that this would not change the conclusions of the paper, but indeed yields more robust results. Hence the revised manuscript now includes analyses on the last 10 years of 30 years long simulations for the fast adjusting metrics.

7.  Page 12: The rationale for the approach to separating the mean and eddy heat fluxes is not clear. A three month average doesn't seem sufficient to separate mesoscale eddy timescales from the mean—why not use an average over the full four years available? Also, considering that resolved eddies still play a role in the R1 simulation, why are the effects of these not also diagnosed and added to the GM contribution?

    The suggested approach would absorb seasonality into eddy heat fluxes, which is what we want to avoid. Nevertheless we tested it and did not find that it changes the conclusions regarding the mean meridional heat transport and adopted the suggestion in the revised version. We agree that the effect of resolved eddies for R1, although only relevant at low latitudes, should be included. Hence we revised the diagnostics to include the combined meridional heat fluxes of GM and the resolved eddy contribution for the R1 experiment.

**Technical corrections**

1.  Remove indent on line following equation (5).

    Done.

**References:**

Marques, G. M., Loose, N., Yankovsky, E., Steinberg, J. M., Chang, C. Y., Bhamidipati, N., ... & Zanna, L. (2022). NeverWorld2: An idealized model hierarchy to investigate ocean mesoscale eddies across resolutions. *Geoscientific Model Development*, *15*(17), 6567-6579.

Munday, D. R., Johnson, H. L., & Marshall, D. P. (2013). Eddy saturation of equilibrated circumpolar currents. *Journal of Physical Oceanography*, *43*(3), 507-532.

---

## Author Comment (AC2)

**Reply to Referee comment #2:**

The authors would like to thank the referee for their time, as well as their invaluable comments and suggestions. In the following each comment, suggestion or concern is replied in green font. Specific revisions of the text are in quotes, with the respective changes highlighted in bold.
* * *
Overall I found this to be a very relevant and useful study, with excellent figures and well-written text. Please see my minor comments below:

Introduction first paragraph: First sentence — there isn't really a scale separation between "underlying processes" and "changes in Earth's climate", there are changes and dynamics on a continuum and they're all linked/interact with each other.

Indeed ! We rephrased the first sentence in : "The vast range of spatial and temporal scales of Earth's climate system and the underlying processes involved, make numerical climate simulations a computationally costly endeavor : **it requires representing the effect of small scales (< 100km) in long simulations (> 500 yr)**."

Second sentence "their" isn't obviously grammatically related to "climate simulations".

Revised version : "Limited available computational resources therefore impose constraints on the horizontal resolution **of future climate projections."**

Line 16: There's a difference between mesoscale eddies and geostrophic turbulence (the latter is a broader term); I'd say something like "mesoscale eddies are the most salient feature arising from geostrophic turbulence".

Thank you for the clarification. Here is the revised line : "These grid scales coincide with the horizontal scale of geostrophic turbulence (Chelton et al., 1998). **A prominent feature associated with turbulence at these scales is the formation of mesoscale eddies."**

Introduction second paragraph: It's not just winds sustaining the PE reservoir but also heterogeneous buoyancy forcing.

Indeed, hence we revised in : "Winds inject kinetic energy at the surface, and, **together with heterogeneous buoyancy forcing,** they sustain a reservoir of potential energy (PE) at large scales."

Line 19: There's a cascade of energy into the first baroclinic mode as well, so it's an upscale and downscale cascade of energy (see Smith and Vallis 2001 Fig. 4 for example). In the barotropic mode there's an upscale transfer, but in the higher modes the energy transfers go both ways and funnel energy into the 1st baroclinic mode.

Thanks for this clarification. revised version "The thereby excited baroclinic modes and nonlinear interactions between them lead **to energy transfers across scales."**

How valid are the parameter values chosen for the linear EOS when considering high-latitude vs. low-latitude behavior (where S vs. T are respectively more dominant in setting density)?

This was extensively explored in Caneill et al. (2022), where they tested this EOS across a range of parameters to investigate what sets the polar transition zone (transition from temperature controlled alpha ocean to salinity controlled beta ocean). We chose the EOS parameters for DINO after personal exchange with Romain Caneill and Fabien Roquet and refer to their studies for reference.

Can you explain more what is meant by the NW2 style bathymetry introducing an "undesirable separation into two basins with respect to dense water formation and overturning"? The real ocean does have this feature so I'm not sure where this hypothesis came from. We found that the ridge was important to setting some of the vertical structure properties of the eddies and potentially the broader circulation (Yankovsky, Zanna, Smith, 2022).

The DINO configuration as presented in the paper is the outcome of a very large number of sensitivity experiments, where we tested the bathymetry, amongst other characteristics. We started with the NW2 style bathymetry including a mid-Atlantic Ridge and found that it splits the subpolar gyre, leading to convection in the eastern half of the basin. The real ocean has a mid-Atlantic ridge indeed, but it is fractured and the Atlantic geometry allows for a coherent subpolar gyre across the basin with deep-convection in the western side of the gyre (the Labrador Sea). We tested opening the ridge in the north (and south, by the way) to mimic this, but concluded that it greatly complicates the interpretation of the results with little benefit for the purpose of testing mesoscale parameterizations. Hence we decided to pursue this study with no ridge. It remains optional in the namelist.

There is no mention of the dissipation scheme being used until Table 2, I recommend stating this in the model equations/setup. In NW2 we had to think at length about a viscosity scheme that could be applied in a consistent way across resolutions (ended up using biharmonic Smagorinsky). This isn't being done here, the viscosity parameterizations in R1 are different in formulation than R4 and R16; could the authors speak more about the reasons and implications of this?

This was done deliberately, as subgrid parameterizations and their numerical schemes should ultimately be the users choice, depending on the aim of the study. Here we provide reference experiments for the DINO configuration, with the associated parameter choices. For R1 we use a less scale selective, more dissipative Laplacian viscosity operator for numerical stability and the GM parameterization, as is commonly done in models of similar horizontal resolution. NW2 is not presented for such coarse resolutions, presumably for the same reasons.

Lines 171-179: Is GM by default added to the higher resolution simulations as well, just with a lower coefficient? I would be more explicit about this. This is in itself a "parameterization" choice that may conflict with other choices the users make on top of that to test other eddy parameterization schemes. For example, in my work on backscatter parameterizations, I found that backscatter can replace the need for GM in eddy permitting simulations (and using the two simultaneously is problematic, see Yankovsky et al. 2024).

GM is deactivated entirely in R4 and R16, for the same reasons you mentioned. This should be clear from the text and we propose to clarify it in the revised version of the paper : "In eddy-permitting and eddy-resolving horizontal resolution, we assume this process to be at least partially resolved **and consequently omit the GM parameterization**."

Line 186: Can you show a figure verifying that the tracers have reached a quasi-equilibrated state? This can be incorporated as a panel into one of the first several figures. I'm curious what is meant by "quasi" here, are the tracers in the deep ocean still evolving? How far out of equilibrium are the higher-resolution simulations? Would be helpful to visualize this in a figure as well. In the higher-resolution simulations, it would be helpful to have more discussion of what the lack of equilibration can introduce error-wise into the analysis.

We have prepared a figure for the revised version that illustrates equilibration of thermodynamic properties : a time-series of the volume of water masses defined by $\sigma_2 < 27$ and $27 < \sigma_2 < 26$. This figure is added in appendix, to illustrate what is meant by "quasi-equilibrium" : surface energetics and main ocean currents are equilibrated, yet there remains a drift in the thermodynamic properties at depth.

How do you propose accounting for the unresolved submesoscale dynamics? Is there any parameterization for those effects implemented, and how might that conflict with the mesoscale parameterization?

We did not include submesoscale parameterizations in DINO, as we aim to assess parameterizations targeting the eddy-permitting regime primarily. NEMO has a parameterization for mixed-layer eddies implemented (Fox-Kemper et al. 2008), which could be activated in the future. As this is outside the scope of this paper, we do not include dedicated experiments. We agree that this choice is arguable : submesoscale parameterizations may indeed interfere with mesoscale parameterizations. We added a sentence about this in the discussion section of the revised paper.

Might be interesting to consider referencing some of the recent work being done on the Oceananigans model in light of the more traditional modeling efforts/studies addressed here. One can make the argument that rather than layering more complex parameterization schemes on top of each other, we should instead focus on developing modeling frameworks that are able to resolve down to submesoscales through GPU-based architectures. See Silvestri et al. 2025 (**https://doi.org/10.1029/2024MS004465**)

We agree with the reviewer, and for this reason we approached Simone Silvestri and the Oceananigans team about 1 year ago. We contributed to implementing a prototype of DINO in Oceananigans (https://github.com/simone-silvestri/WenoNeverworld.jl/blob/f8539306879e357d3527fbeda51b14cf2c126c67/dino/WenoDINO.ipynb). However, these developments are not ready yet for this publication.

**References:**

Caneill, R., Roquet, F., Madec, G., & Nycander, J. (2022). The polar transition from alpha to beta regions set by a surface buoyancy flux inversion. *Journal of Physical Oceanography*, *52*(8), 1887-1902.

Fox-Kemper, B., Ferrari, R., & Hallberg, R. (2008). Parameterization of mixed layer eddies. Part I: Theory and diagnosis. *Journal of Physical Oceanography*, *38*(6), 1145-1165.